# Fracture Evolution Characteristics and Deformation Laws of Overlying Strata during the Initial Period of Longwall Mining: Case Study

**Chuantian Li [1,2], Yongliang He [1,2,*], Xiaoyuan Sun [1] and Yuping Fu [1]**

[1]  School of Engineering for Safety and Emergency Management, Taiyuan University of Science and Technology, Taiyuan 030024, China; 2009013@tyust.edu.cn (C.L.); 2011039@tyust.edu.cn (X.S.); 2011014@tyust.edu.cn (Y.F.)

[2]  Intelligent Monitoring and Control of Coal Mine Dust Key Laboratory of Shanxi Province, Taiyuan University of Science and Technology, Taiyuan 030024, China

*  Correspondence: hylust@163.com

**Abstract:** Coal mining causes overlying rocks to collapse and be destroyed. Overburdened crevasses provide a channel for gas discharge, which is a serious safety hazard. To study the evolution characteristics and migration pattern of overburdened fissures during the initial mining period, the 24207 working face of the Shaquan mine was used as a research object. Through similar physical simulation tests, a mechanical model of the mining structure during the initial mining period was constructed to explore the factors influencing the movement pattern of the overburden strata during the initial mining period. The research results show that the evolution of mining-induced fractures in the overburdened strata in the initial mining period mainly experience the slow and rapid rising stages of the fracture dimension, while the stable mining period is in the stable development stage of the fracture dimension. The research results will help supplement and improve the theory of gas disaster prevention and comprehensive resource utilization in the initial mining stage under the mining conditions of high gas and low permeability coal seam group, achieve the goal of "coal and gas co-mining", and ensure the safe and efficient production of mines.

**Keywords:** longwall mining; overburden fractures; gas hazard; physical simulation; initial mining period

## 1. Introduction

Coal is a basic energy source and an important raw material for China's industrial production and economic development [1,2]. For many years to come, coal will remain the main energy source and strategic core of China for the foreseeable future. The coal mine safety situation is still not optimistic because the key areas that determine the unstable development of coal mine safety and its deep-seated contradictions have not been effectively resolved [3]. For example, coal mining causes overlying rocks to collapse and be destroyed. Overburdened crevasses provide a channel for gas discharge, which causes a serious safety hazard [4–6]. Therefore, there is an urgent need to study the mining-induced fracture evolution and overburden deformation laws during the initial period of longwall mining.

The initial mining period is defined by some scholars as the advancement time of the mining distance from the beginning of the seam extraction to the initial weighting of the basic roof and the unstable gas emissions. The development height of mining-induced fractures in the normal mining process of the mine face can reach 145 m, which is obviously difficult to reach in the initial mining period [7,8]. Zhang [9] studied the complex differences between the mining process and its related static and dynamic geostress fields, further characterizing the volume characteristics of the mining fracture system by scanning fractured coal samples with an industrial computer tomography system. Wenli Yao [10] used three-dimensional particle flow code (PFC3D) software to simulate the development of the fracture network in the Ganzhuang Coal Mine. There are many

periodically connected fractures in the overburden. Shengwei Li [11] used universal distinct element code (UDEC) software to simulate the spatial distributions of the stress field and fracture field of three typical underground mining layouts: pillar-free mining, top coal caving mining and protective coal seam mining. Zetian Zhang [12] conducted laboratory experiments, computed tomography (CT) scanning and image analysis on the three-dimensional fracture system of a coal body concerning the stress conditions induced by three typical mining methods: top coal caving mining, pillar-free mining and protective coal seam mining. Hui Zhuo [13] conducted similar simulations indoors, observed the cracks induced by ground mining, and detected the air leakage of cracks. V. Palchik [14] analyzed the main factors affecting the large aperture range of horizontal fractures caused by traditional longwall mining in the Torezko Snezhnyanskaya coal area in Ukraine and found that the main factors affecting the opening of mining-induced horizontal fractures are the thicknesses of the mining coal seams, the distances between the mining coal seams and the horizontal fractures, the uniaxial compressive strengths of the rock layers above and below the fractures, the thicknesses of the rock layers above and below the fractures, and the thicknesses of the key layers. Cao Jie [15] used the discrete element software UDEC and the multiphysics software COMSOL to simulate the gas migration of mining-induced fractures above a goaf. With the advancement of the working face, the overlying rock of the goaf gradually forms a trapezoidal mining-induced fracture network, and the size of the fracture network increases with increasing working face distance. Jianguo Zhang [16] studied the distribution and evolution laws of the overburden of the fractured zone after the excavation of the J15 coal seam in the Pingdingshan No. 10 Mine by combining similar and numerical simulations and conducted a numerical simulation of the gas migration law in the fractured zone. Gao [17] studied the abutment pressure, complexity and connectivity evolution trend of mining-induced fracture networks under high pressure relief conditions. According to the evolution speeds of the mining fractures and the abutment pressure in the mining face, the evolution laws of fractures and the relationship between connectivity and abutment pressure under the top coal caving condition were preliminarily studied [18–20].

To date, research on key mining strata deformation, overburden migration, load transfer and mining fracture evolution has changed from macroresearch on the whole process to microresearch on local and staged mining [21–25]. The descriptions of the deformation characteristics of the overlying strata under mines and the laws of gas emissions are mainly focused on the normal mining stage [26–29]. The study of the decisive mechanisms of mining stress transfer and surrounding rock fissure evolution on abnormal gas emissions in the special stage of initial mining must be further deepened [30,31]. By taking the 24207 working face of the Shaqu Mine as the research object, a similarity simulation experiment was conducted indoors, and the overburdened caving characteristics and the development laws of the mining-induced fracture were analyzed in the initial period of longwall mining.

## 2. Engineering Background

The 24207 working face of the Shaqu Mine of Huajin Coking Coal Co., Ltd., Lvliang, China. belongs to the seventh inclined longwall working face in the North No. 2 mining area. It is estimated that the floor elevation of the working face is +360~+450 m, and the overlying surface is a loess-covered area, with an elevation of +862~+1007 m. Additionally, it is estimated that the mountain cover is hilly, with a thickness of 423~632 m.

The 24207 working face is a combined mining of the #3 + 4 coal seam, with thicknesses of 4.3–4.7 m, an average of 4.6 m, and a dip angle of 4–7°. According to the drilling data near the working face, there is a thin interlayer between the #3 and #4 coal seams. The vertical distances of the #3 and #4 coal seams are 100 mm. The average distance between the #3 + 4 coal seam and the overlying #2 coal seam is approximately 10.5 m, and the average distance between the #3 + 4 coal seam and the overlying #5 coal seam is approximately 5.6 m. The average thicknesses of the #2 and #5 coal seams are 1.04 m and 3.3 m, respectively.

There is 0.2 m of mudstone in the partial pseudo roof of the 24207 working face, which is not developed. The direct roof of the #3 + 4 coal seam is grey-black, medium-fine sandy mudstone, with a thickness of 5.5 m and a Proctor hardness f of 5, indicating that the roof is hard, brittle and easy to fall. The main roof is greyish-white medium sandstone, 5.59 m thick and mainly composed of quartz, followed by feldspar, and has uniform bedding and has a thick layer. The direct bottom is grey medium sandstone with Proctor hardness $f = 5$, which is a basically stable rock stratum. The old bottom is 2.5 m of black siltstone. The comprehensive stratigraphic histogram of the 24207 working face is shown in Figure 1.

| Depth of stratum/m | Histogram | Rock name | Rock description |
|---|---|---|---|
| 7.30 | | Medium sandstone | Gray white medium sandstone, mainly quartz, followed by feldspar, uniform bedding. |
| 2.07 | | Mudstone | Black mudstone. |
| 1.04 | | 2# coal | 2 # coal seam, semi bright briquette, powder. |
| 1.75 | | Carbonaceous mudstone | Black carbonaceous mudstone, containing plant fossil fragments. |
| 1.61 | | Fine sandstone | Gray fine sandstone, medium thick layered, followed feldspar, uniform bedding. |
| 4.50 | | Medium sandstone | Gray white medium sandstone, mainly quartz, followed by feldspar, uniform bedding. |
| 1.09 | | Siltstone | Dark gray siltstone, thin layer, vein bedding. |
| 5.50 | | Sandy mudstone | Gray black mudstone, containing plant fossil fragments, with siderite. |
| 4.17 | | 3+4# coal | Semi bright briquette, glassy luster, with internal cracks. |
| 1.10 | | Medium sandstone | Gray white medium sandstone, mainly quartz, followed by feldspar, uniform bedding. |
| 2.50 | | Siltstone | Black siltstone with fossil plant fragments. |
| 2.00 | | Mudstone | Black mudstone. |
| 3.30 | | 5# coal | Semi bright briquette, glassy luster, with internal cracks. |
| 2.60 | | Sandy mudstone | Gray black mudstone, a large number of plant root fossils can be seen. |
| 1.70 | | K3 sandstone | Brown gray coarse sandstone, argillaceous cementation. |

**Figure 1.** Comprehensive histogram of the coal and rock layers on the 24207 working face.

The 24207 working face is equipped with three main roadways—a track chute, a return air chute and a belt chute—and 28 crossovers are constructed between the return air chute and the belt chute. The return air duct is 1514 m long, the belt duct is 1613 m long, and the track duct is 1691 m long. To conduct the relevant experimental research on the initial mining stage of the mining face, the length of the working face within the initial 885 m range from the open cut is 220 m, and then the working face is extended to 260 m. The minable strike length of the working face is 1548 m, and the minable area is 0.34 km². The mining method involves full-height mining and inclined longwall retreating mining that is fully mechanized with full caving. The coal seam thicknesses of the working faces are 4.3–4.7 m, with an average thickness of 4.6 m; thus, the average mining height of the working face is set as 4.5 m. The working face layout is shown in Figure 2. The No. 2 coal seam is located in the upper part of the Shanxi Group and is the most stable and mostly mineable coal seam in the entire wellfield. The coal seam has a high gas content, and the mine has a high gas gush (relative gas gush 9.10 to 26.83 m³/t, average 17.97 m³/t; absolute gas gush 15.80 to 46.58 m³/ min, average 31.18 m³/ min). The lithology surrounding Coal Seam No. 2 is mainly carbonaceous mudstone, mudstone and sandy mudstone, followed by fine-grained sandstone and siltstone. After the mining of the working face, the free caving method is used to treat the goaf, and the untreated goaf is located behind the working face. The mudstone is generally thick (10.12–27.18 m, average 16.67 m), of generally good integrity and is dense and hard. The ability of the surrounding rocks of the No. 2 coal seam to cap the gas is good, and the gas is easily preserved and enriched for reservoir formation, which is one of the main geological factors contributing to the generally high gas content of the seam.

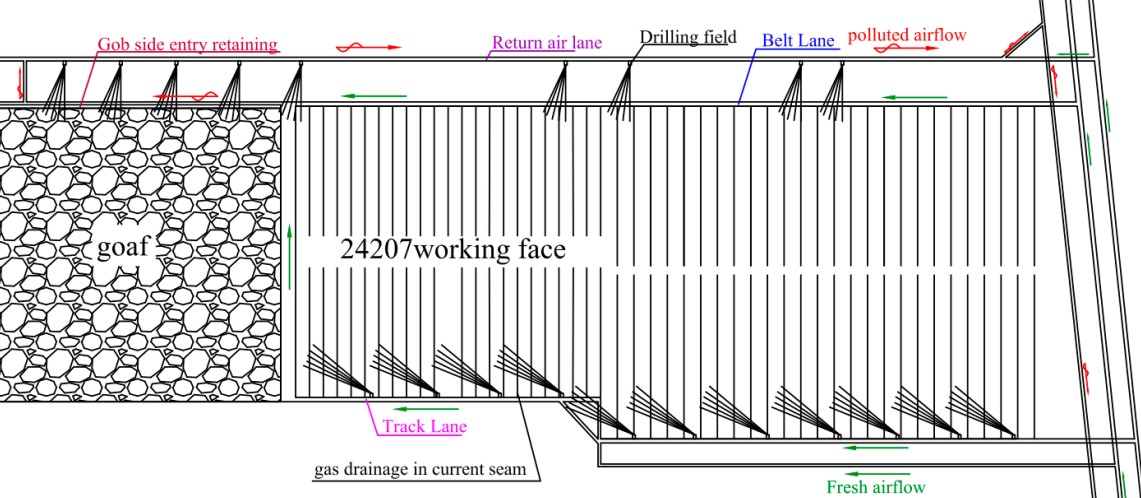

**Figure 2.** Layout of the 24207 working face.

## 3. Research on Similar Simulations of the Overburden Collapse Law

### 3.1. Similar Simulation Experimental Design

Combined with the research purpose, model reality and existing test conditions, this test uses a self-developed two-dimensional physical similarity simulation test platform and a plane stress model for test research. The effective size of the similar model test platform is length × width × height = 1850 mm × 200 mm × 2500 mm, which is composed of multiple components, including a hydraulic loading system, model frame, channel steel and plexiglass side guard board, with #20 channel steel wrapped around and at the back side, 25-mm-thick plexiglass board at the front side, and free/loading ends at the top of the model [32–34].

A similarity simulation experiment is a laboratory research method based on similarity theory and dimensional analysis [35–37]. Referring to the geological and mining technical conditions of the simulation prototype in the previous section, the simulation strike length ($l_m$) is 111 m. Since the effective dimension length of the test model ($l_p$) is 1850 mm, the geometric similarity ratio ($C_i$) is as follows:

$$C_i = \frac{l_m}{l_p} = \frac{1.85}{111} = \frac{1}{60} \tag{1}$$

According to experience, in general, the average unit weight of the primary coal and rock mass ($\gamma_m$) is 2.5 t/m³, and the unit weight of the model ($\gamma_H$) is 1.7 t/m³; thus, the unit weight similarity ratio calculated ($C_r$) in the test is as follows:

$$C_r = \frac{\gamma_H}{\gamma_m} = \frac{1.7}{2.5} = 0.68 \tag{2}$$

When the geometric similarity ratio $C_i$ and bulk density similarity ratio $C_r$ are determined, the strength similarity ratio $C_a$ is determined. The calculation results are as follows:

$$C_a = C_i \cdot C_r = \frac{0.68}{60} = 0.01133 \tag{3}$$

### 3.2. Proportions and Calculations of Similar Materials

The unit weight and compressive strength values of the simulated materials are converted according to the stress ratio, and the materials are selected in a layer-by-layer manner by referring to many mix ratio test data [38–40]. The mechanical properties of the roof and floor strata of the simulated mining coal seams and their corresponding proportions are shown in Table 1.

**Table 1.** Main parameters and material ratio of the surrounding rock and coal seam for simulated mining.

| Serial | Lithology | Compressive Strength/KPa | Layer Thickness/cm | Density/(kg·m$^{-3}$) | Proportioning | River Sand/kg | Cement/(Lime) kg | Gypsum/kg | Water/kg | Borax/g |
|---|---|---|---|---|---|---|---|---|---|---|
| 1 | Medium sandstone | 339 | 12.17 | 1768 | 337 | 117.0 | 48.6 | 113.5 | 23.2 | 463.1 |
| 2 | K-sandstone | 652 | 2.8 | 1768 | 655 | 10.9 | 3.7 | 3.7 | 1.8 | 18.3 |
| 3 | mudstone | 369 | 3.45 | 1219 | 973 | 22.9 | 6.8 | 2.9 | 9.7 | 96.9 |
| 4 | #2 coal seam | 420 | 1.73 | 925 | 537 | 8.7 | 9.58 | 22.3 | 6.4 | 63.6 |
| 5 | Carbonaceous mudstone | 411 | 2.9 | 1713 | 855 | 14.7 | 1.88 | 1.8 | 1.8 | 18.4 |
| 6 | Fine sandstone | 576 | 2.7 | 1754 | 555 | 8.8 | 4.4 | 4.4 | 1.8 | 17.5 |
| 7 | Medium sandstone | 339 | 9.2 | 1768 | 337 | 18.1 | 12.6 | 29. 5 | 6.0 | 12.0 |
| 8 | Sandy mudstone | 311 | 9.2 | 1678 | 436 | 22.8 | 10.3 | 23.9 | 5.7 | 57.1 |
| 9 | #3 + 4 coal seam | 89.25 | 7.7 | 938 | 473 | 10.7 | 11.2 | 4.8 | 2.7 | 53.47 |
| 10 | Medium sandstone | 339 | 1.8 | 1768 | 337 | 3.5 | 2.5 | 5.8 | 1.2 | 23.5 |
| 11 | Siltstone | 669 | 4.2 | 1744 | 955 | 24.4 | 1.4 | 1. 4 | 2.7 | 27.1 |
| 12 | Mudstone | 369 | 3.3 | 1700 | 973 | 18.7 | 1.5 | 0.6 | 2.1 | 20.8 |
| 13 | #5 coal seam | 89.25 | 5.5 | 925 | 473 | 7.5 | 7.9 | 3. 4 | 1.8 | 37.6 |
| 14 | Sandy mudstone | 311 | 4.3 | 1677 | 436 | 8.0 | 4.8 | 11.2 | 2.7 | 26.7 |

### 3.3. Experimental Scheme

The length of the similar model is 1850 mm. According to the geometric similarity ratio of 1/60, the corresponding actual excavation length is 111 m. Through theoretical analyses and field tests, the model can meet the research goal in the initial mining period. The thickness of the model is 70.95 cm, corresponding to the actual thickness of 42.57 m. According to the exploration drilling conditions of the 24207 working face, it is estimated that the floor elevation of the working face is +360~+450 m, the overlying surface of the working face is a loess-covered area, the ground elevation is +862~+1007 m, and the estimated thickness of the cover mountain is 423~632 m. By taking the average of the two values, the simulated mining depth is approximately 500 m, the simulated rock thickness is 43 m, and the compensation load to be applied is as follows:

$$q = \gamma(H - h) = 2.5 \times (500 - 43) = 1142.5 \text{t/m}^2 = 11.4 \text{MPa} \tag{4}$$

$$q' = C_\alpha \times q = 0.01133 \times 11.2 = 0.1295 \text{MPa} \tag{5}$$

After preparing the model, it stands for 3 days to fully ensure the stability and dryness characteristics of similar materials. Then, the bottom channel steel of the front back plate of the model is removed, and a mining test is conducted from right to left at the #3 + 4 coal seam. Before formal excavation, an open cut with a width of 13.33 cm (8 m compared with the actual) and a height of 7.7 cm (4.62 m compared with the actual) is first constructed, and then a formal simulation is conducted with the actual site and similar parameters, as shown in Figures 3 and 4.

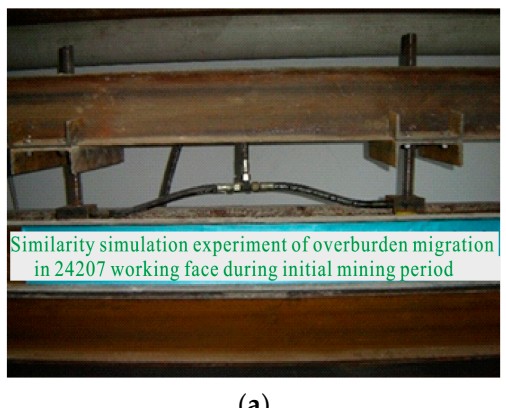
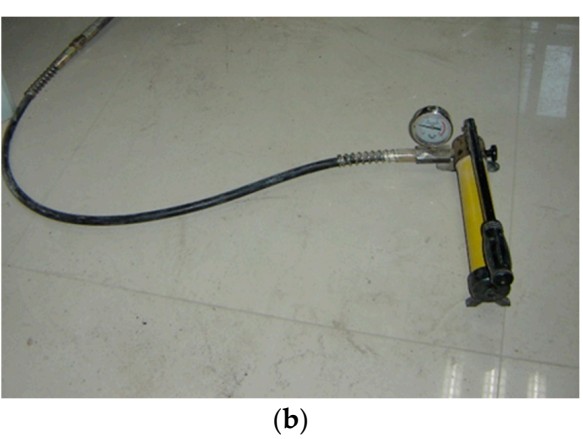

(**a**)                              (**b**)

**Figure 3.** Stress loading device for the 24207 working face simulation model. (**a**) Top loading device of a similar model. (**b**) Hydraulic unit manual loading pump.

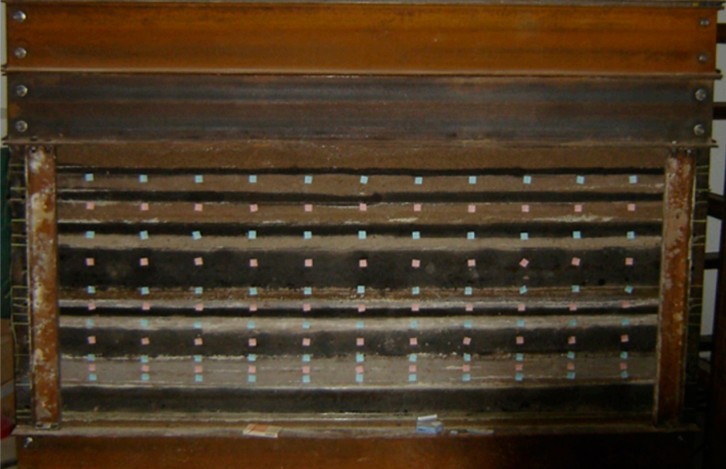

**Figure 4.** Similarity simulation test device.

To fully explore the deformation law of the overlying strata in the initial mining stage of the working face, two groups of models are built for comparative simulation, which are defined as Model I and Model II. The coal and rock settings of the two groups of models are identical, and 8 m protective coal pillars are set on both sides of the mining seam to reduce the impacts on the overlying rock stress and the stable structures of coal and rock. The differences between the two models mainly include two aspects: ① the mining advancement speed is different, in which Model I is set according to the actual mining speed (the first month's footage is 24 m, the second month's footage is 39 m, and the third month's footage is 54 m), and Model II selects the average of the above monthly footage (−39 m) for simulation, and ② the mining direction is different. Model I adopts the excavation method from left to right, and Model II adopts the excavation direction from right to left [41–44].

*3.4. Model Measurement Point Setting*

To fully discuss the deformation laws of mining dynamic load transfer and fracture evolution in the initial mining period, several stress measuring points are arranged in a similar model test; these points are supplemented by stress sensors, static and dynamic strain gauges and a computer automatic data acquisition system to measure the model stress. Additionally, in the model, the displacement dial indicator (dial indicator) is used with the cross layout method to observe the displacement changes in the overlying strata on the roof, and the graphic sketch and photography methods are used to reflect the strata collapse migration and balance structure formation during the initial mining stage.

(1)    Layout and testing method of rock stress measuring points

During the excavation of the working face, the coal and rock masses that originally bear the stress of the overlying strata are mined, causing the load to transfer to the surrounding area and leading to uneven stress distribution. To study the law of stress migration, the BE120-3AA resistance strain gauge is pasted on the square rubber, and the BM-2B static strain gauge is externally connected. The stress value of each key point is monitored with the calibration value of the strain root. During the test, 11 rows of resistance strain gauges are arranged in the horizontal direction in the coalbed, with a spacing of 15 cm. Three rows are arranged in the vertical direction at 7 cm, 15 cm and 35 cm above the mining layer. The layout of the stress measuring points is shown in Figure 5.

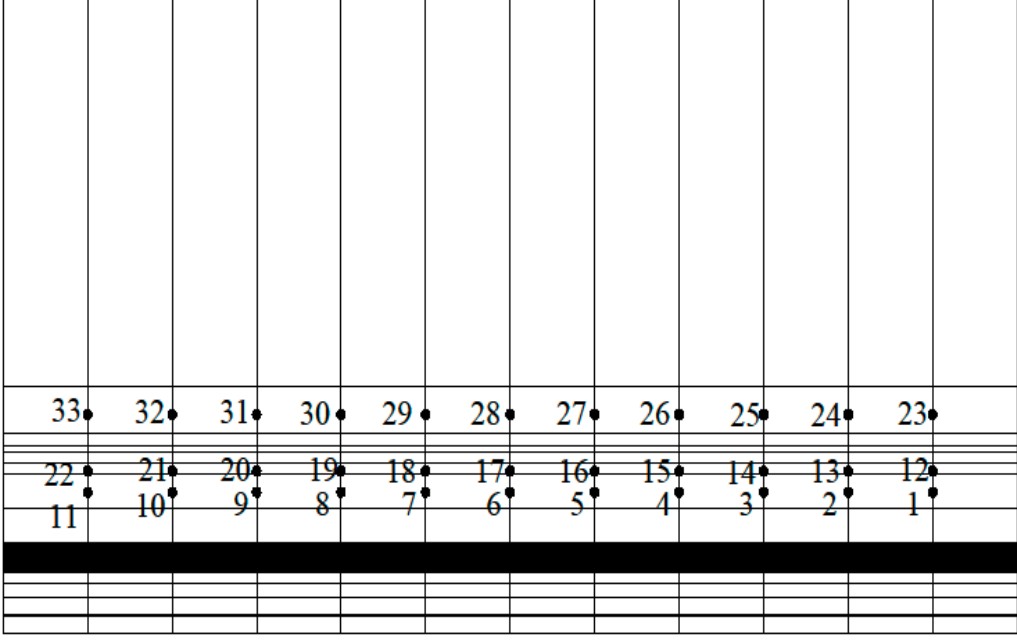

**Figure 5.** Schematic diagram of stress measuring points arrangements.

(2)    Layout and testing method of displacement measuring points

The reasonable arrangement of the displacement measuring points is key for accurately analyzing the evolution laws of overlying rock fractures in the initial mining stage. For this reason, 11 rows of measuring points are arranged horizontally above the #3 and #4 coal seams, with a spacing of 15 cm. In the vertical direction, 121 displacement measuring points are arranged from the bottom to the top from 3 cm above the coal seam. The first three rows of the bottom spacing are 3 cm, the middle four rows are 5 cm, and the upper four rows are 20 cm, as shown in Figure 6.

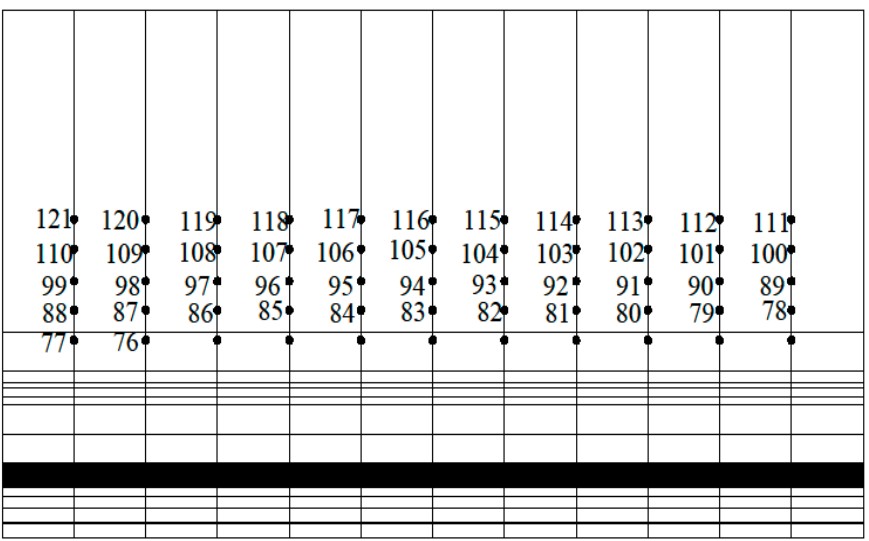

**Figure 6.** Schematic diagram of displacement measuring points arrangement.

## 4. Failure Characteristics of Overlying Strata

### 4.1. Analysis of the Direct Roof Caving Process

According to the pillar analysis and mining simulation test results of the floor rock strata of the #3 + 4 coal seam in the 24207 working face of the Shaqu Coal Mine of Huajin Coking Coal Co., Ltd., Lvliang, China, the sandy mudstone above the #3 + 4 coal seam is a direct roof with a total thickness of 5.5 m, and the model height is 9.15 cm. In the similarity simulation, to facilitate analysis, the direct roof is directly laid in three layers at the top, which are defined as the first, second and third layers from bottom to top. The height of each layer is 3.05 cm.

When the working face advances to 26.98 m, the direct roof of the #3 + 4 coal seam is separated for the first time, and transverse fractures between layers begin to form, as shown in Figure 7a. When the working face advances to 28.78 m, the layer separation phenomenon becomes increasingly obvious, the horizontal cracks gradually expand upwards, and vertical cracks still do not appear, as shown in Figure 7b. When the working face advances to 31.2 m, vertical cracks begin to appear, and the interlayer transverse cracks extend and slightly bend at the direct roof rock layer 2 m above the working face, as shown in Figure 7c. When the working face advances to 35.97, the first and second layers of the direct roof fall simultaneously for the first time, and the irregular rock block presents a layered structure. The third layer of the direct roof is affected by overburden pressure and subsidence deformation, and it maintains relatively good integrity despite slight bending, as shown in Figure 7d. When the working face advances to 37.2 m, under the action of dead weight and overburden pressure, the first collapse of the third layer of the direct roof occurs. At this time, the overall collapse thickness is 5.5 m, as shown in Figure 7e. At the initial stage, the collapsed rock strata cannot fully fill the goaf, and there is still 1.3 m of activity space in the upper part. In addition, the direct roof is completely collapsed, and the basic roof of the upper 19.3 m is exposed simultaneously. When the working face is pushed forwards to 39 m, a 19.0 m long direct roof collapses again, and the collapse of the

direct roof forms a bridge type (arch supported) structure; that is, one side is above the working face, and the other side is on the floor of the goaf, as shown in Figure 7f. When the working face advances to 42 m, the direct roof collapses for the third time. The collapse step is 3.0 m by calculation, as shown in Figure 7g. When the working face advances to 47.5 m, the direct roof collapses for the fourth time with a step of 4.5 m. The direct roof rock is basically collapsed due to the fracture of the rear of the support, forming an obvious temporary balance structure of the direct roof, as shown in Figure 7h.

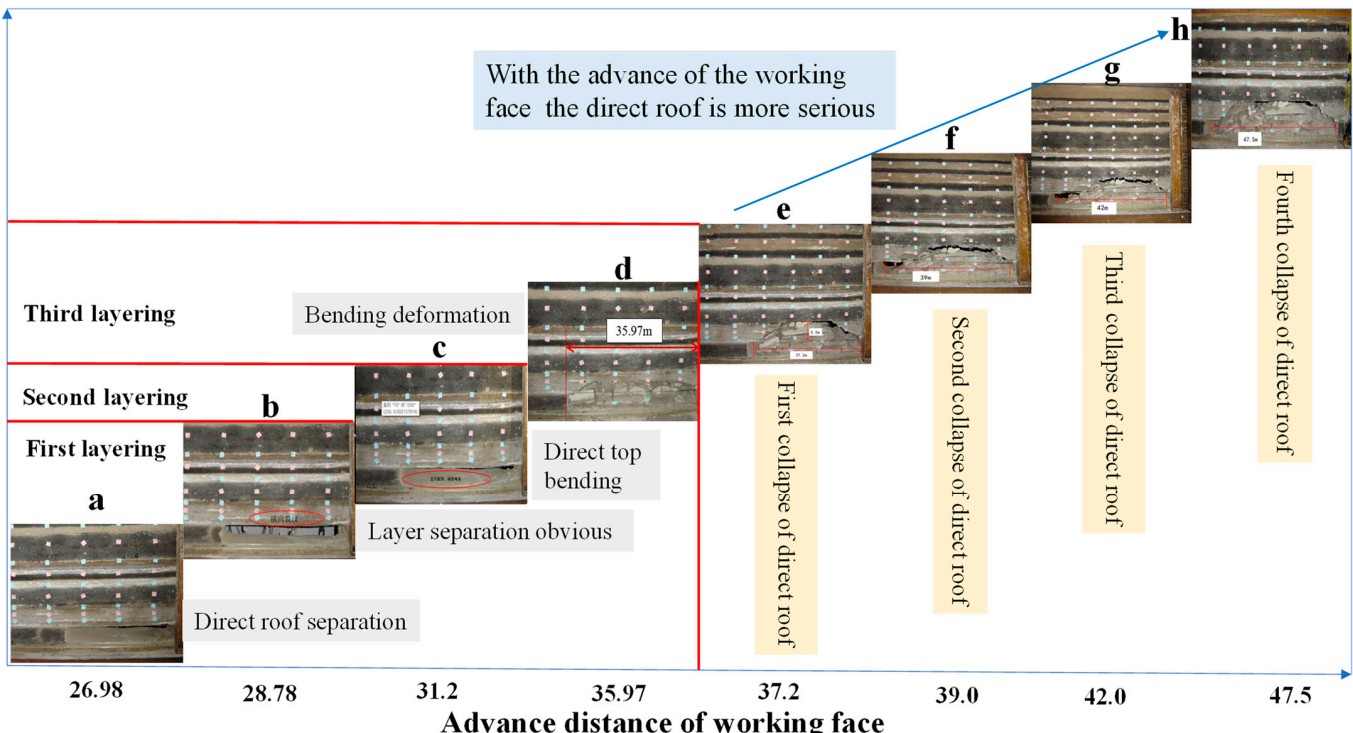

**Figure 7.** Schematic diagram for the collapse process of the direct roof.

### 4.2. Primary Collapse Process of the Basic Roof

When the working face advances to 48.56 m, under the combined effect of the mining operation and the overburden pressure, the first layer of the basic roof begins to show cracks and then collapses with a thickness of 2.75 m, as shown in Figure 8a. When it is pushed forwards to 58.16 m, the first layer of the basic roof with a length of approximately 10 m collapses, and it is covered on the direct roof to form a hinge structure; that is, one end is above the direct roof that is collapsed and compacted, while the other end is still hinged with the basic roof that has not collapsed, as shown in Figure 8b. When it advances to 62.9 m, the second layer of the basic roof slips and loses stability, the overlying strata collapses accordingly, and the ground pressure of the working face appears for the first time. According to this finding, the first weighting step of the basic roof is 62.9 m, and the caving thickness is 5.5 m, as shown in Figure 8c.

### 4.3. Regularity of the Periodic Collapse of the Basic Roof

After the first weighting of the roof of the model test working face, with the continuous advance of mining, the fracture instability movement of the key rock stratum of the stope roof (the basic/old roof) enters a periodic activity stage with a fundamentally different regularity from the initial fracture instability [45].

During the initial weighting of the working face, as shown in Figure 9a, in addition to the phenomena of fracture and collapse from the direct top to the key layer and to the basic roof rock, many vertical fractures are derived in front of the working face, and the evolution of the fractures increase with the progression of the mining operation.

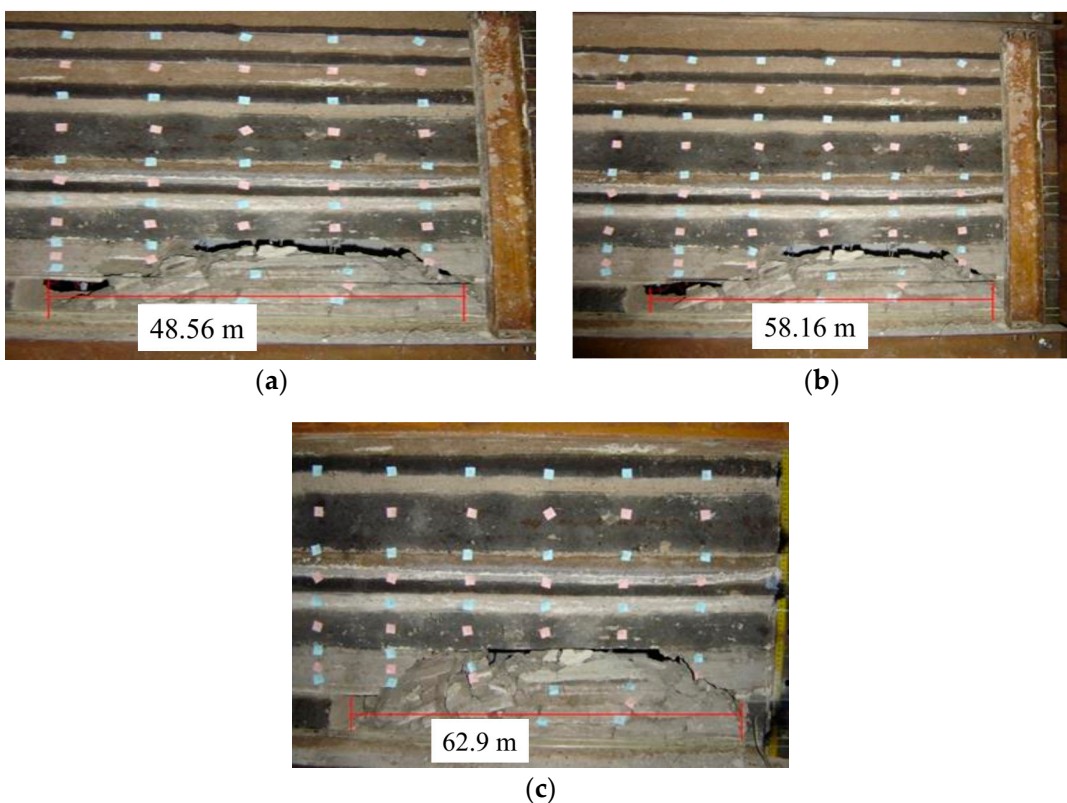

**Figure 8.** Schematic diagram for the subsequent caving process of the basic roof. (**a**) Advance the working face to 48.56 m. (**b**) Advance the working face to 58.16 m. (**c**) Advance the working face to 62.9 m.

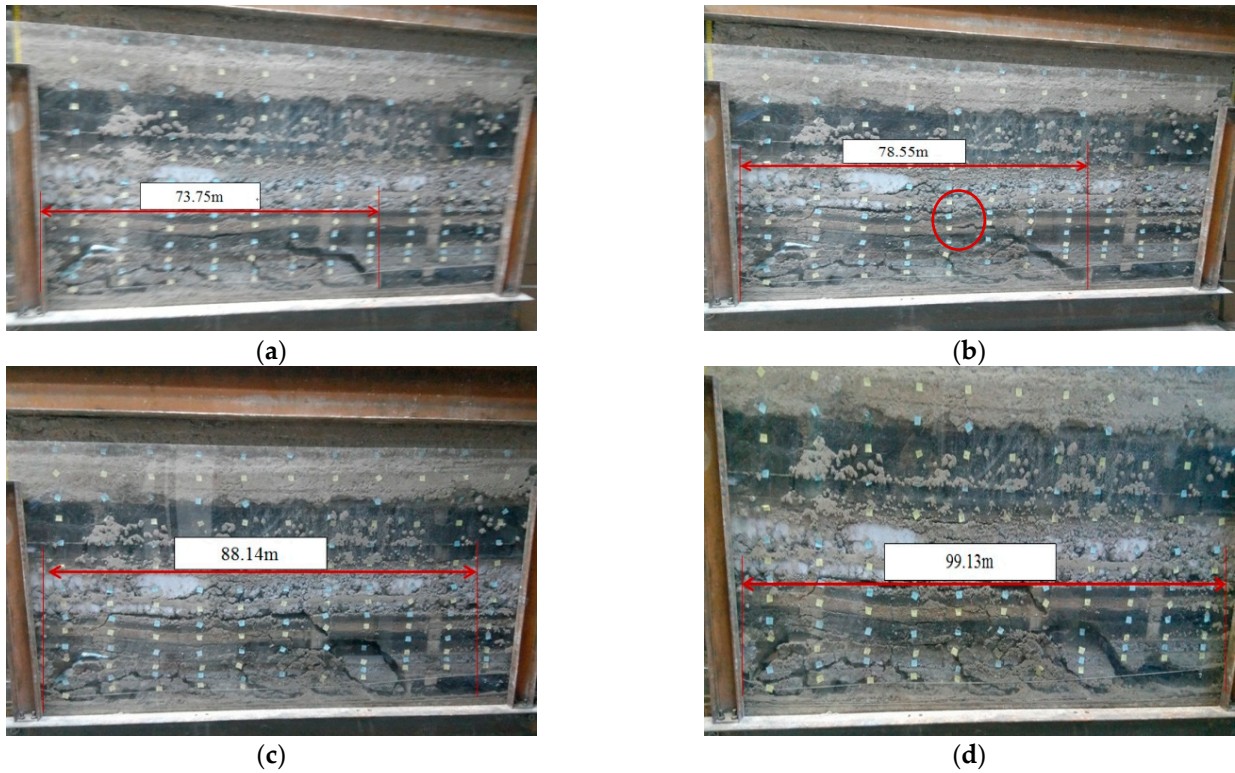

**Figure 9.** Schematic diagram of cyclic pressure. (**a**) First periodic pressure. (**b**) Second periodic pressure. (**c**) Third periodic pressure. (**d**) Fourth periodic pressure.

As shown in Figure 9b, when the working face is mined to 73.75 m, many derived fractures are suddenly connected; the synchronous fracture of the basic roof, which maintains a relatively long suspended state from the fracture location of the initial collapse instability to the coal wall of the working face, leads to the subsequent collapse of its overlying strata, thus causing the first periodic weighting of the working face. The first periodic weighting step is 10.85 m.

At the first periodic weighting, several relatively closed cracks appear in the second layer of the basic roof simultaneously, and the crack expansion speed increases with the advancement of the working face. When the working face advances to 78.55 m, the above cracks finally form a relatively large vertical through crack, which marks the second cycle of compaction.

When the working face continues to advance, the basic roof is suspended again. Because the side of the coal pillar behind the goaf undergoes long periods of loading and deformation, and because the filling degree is relatively poor, the basic roof rock experiences the failure form of tension before shearing along the boundary of the coal pillar. When the working face advances to 88.14 m, the long-distance suspended basic roof in front of the working face collapses, and the third periodic weighting appears, as shown in Figure 9c.

When the working face advances to 99.13 m, the fourth periodic weighting of the basic roof rock occurs, as shown in Figure 9d. The fourth cycle weighting step is 10.99 m. The fractures of the basic roof rock and the overlying rock are not synchronous. The lower part of the basic roof is broken into two parts, while the upper part of the basic roof and the overlying rock still maintain relative integrity and continuity, and the articulated structure of the masonry beam is formed.

*4.4. Correlation between Fracture Evolution and Caving Step Distance in the Mining Process*

During the initial mining period of the longwall working face, the mining fractures in the overlying strata present a double trapezoidal distribution pattern along the advancement direction of the working face. The internal trapezoid refers to the compaction area of the falling rock, while the external trapezoid refers to the crack development area; the crack area on the side near the working face is greater than that on the side near the goaf. The distribution of cracks during the mining period should be that the internal trapezoidal compacted area gradually increases and develops along the rear of the goaf. The external trapezoidal crack development area gradually develops along the horizontal and vertical directions of the working face, and the trapezoidal structure gradually damages, forming saddle-shaped and elliptical throwing body structures. The development ranges of fractures in the overlying strata of the goaf during the initial mining period are larger in the horizontal direction and smaller in the vertical direction than for the normal mining period, and the fracture zone has not yet fully formed. Therefore, the layout of the gas extraction boreholes during the initial mining period is mainly based on horizontal long boreholes.

When the working face advances to 37.2 m, under the action of self-weight and overburden pressure, the first collapse of the third layer of the direct roof occurs; the overall collapse thickness is 5.5 m. In the initial mining stage, the collapsed rock strata cannot fully fill the goaf, and there is still 1.3 m of activity space in the upper part. In addition, the direct roof is completely collapsed, and the upper basic roof is exposed simultaneously. When the working face advances to 39 m, a 19.0 m long direct roof collapses again, which forms a bridging (arch-supported) structure; that is, one side of the roof is placed above the working face, and the other end is placed on the floor of the goaf. When the working face advances to 42 m and 47.5 m, the direct roof collapses for the third and the fourth time, respectively. The first pressure appears when the working face advances to 57 m, at which time the first and second layer of the basic roof collapse synchronously under the combined effect of the mining operation and the overburden pressure.

When the working face is recovered to 68 m, 83.0 m, 91.0 m and 101 m, it experiences four cycles of weighting, and its advancement and step distances are shown in Table 2.

**Table 2.** Step statistics of typical nodes in the mining process.

| Serial | Collapse/Pressure | Advancement of Working Face | | Step Distance | |
|---|---|---|---|---|---|
| | | Model/mm | Prototype/m | Model/mm | Prototype/m |
| 1 | First collapse of direct roof | 620 | 37.2 | 620 | 37.2 |
| 2 | Second collapse of direct roof | 650 | 39 | 30 | 1.8 |
| 3 | Third collapse of direct roof | 700 | 42 | 50 | 3.0 |
| 4 | Fourth collapse of direct roof | 791.7 | 47.5 | 91.7 | 5.5 |
| 5 | Primary pressure of basic roof | 950 | 57 | 950 | 57 |
| 6 | First cycle pressure | 1133.3 | 68.0 | 183.3 | 11 |
| 7 | Second cycle pressure | 1383.3 | 83.0 | 250 | 15 |
| 8 | Third cycle pressure | 1516.7 | 91.0 | 133.3 | 8 |
| 9 | Fourth cycle pressure | 1683.3 | 101.0 | 166.7 | 10 |
| 7 | Average step distance of periodic pressure | | | 183.3 | 11 |

## 5. Overburden Deformation Law

### 5.1. Law of Direct Roof Sinking and Falling

Figure 10 shows that as the working face starts to cut holes, the measuring points on each layer of the direct top appear to slowly separate and sink, collapsing with the continuous progress of the working face. The initial excavation speed is relatively fast, and the progressive collapses of measuring points No. 1~3 do not occur, maintaining a long exposure time and synchronizing layered collapses at 35.97 m.

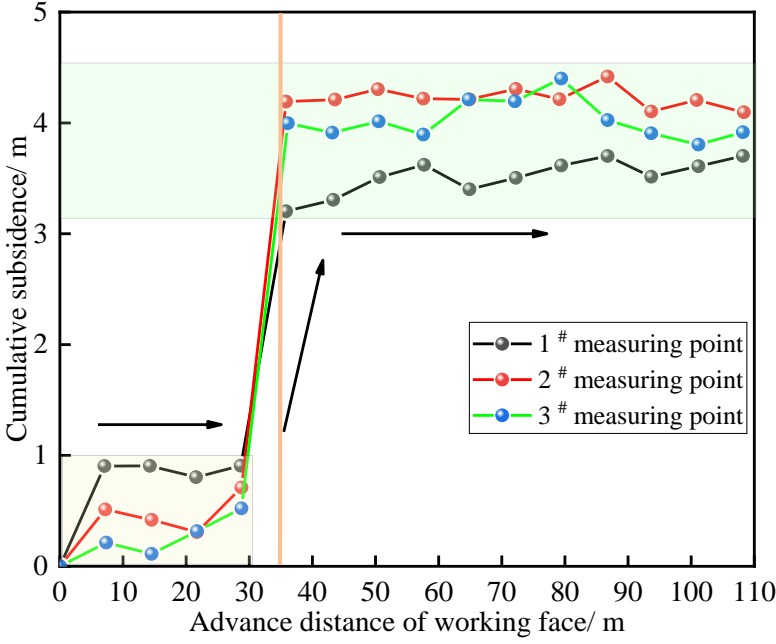

**Figure 10.** Curve of direct roof subsidence and cross fall.

### 5.2. Laws of Basic Roof Sinking and Falling

The displacement change in each measuring point on the basic roof is relatively small at the initial stage, and it gradually sinks due to the layer separation phenomenon, with the maximum subsidence not exceeding 1.5 m. However, when the working face advances to 57 m, measuring points 35 and 36 collapse synchronously, with the maximum subsidence exceeding 4.5 m; the collapse is accompanied by the initial weighting of the stope roof. Different from measuring points No. 35 and No. 36, measuring point No. 34 does not collapse, and its subsidence is basically maintained at 1.5~2.0 m. Obviously, this phenomenon is caused by the hinge structure of measuring point No. 1 at its bottom part, as shown in Figure 11.

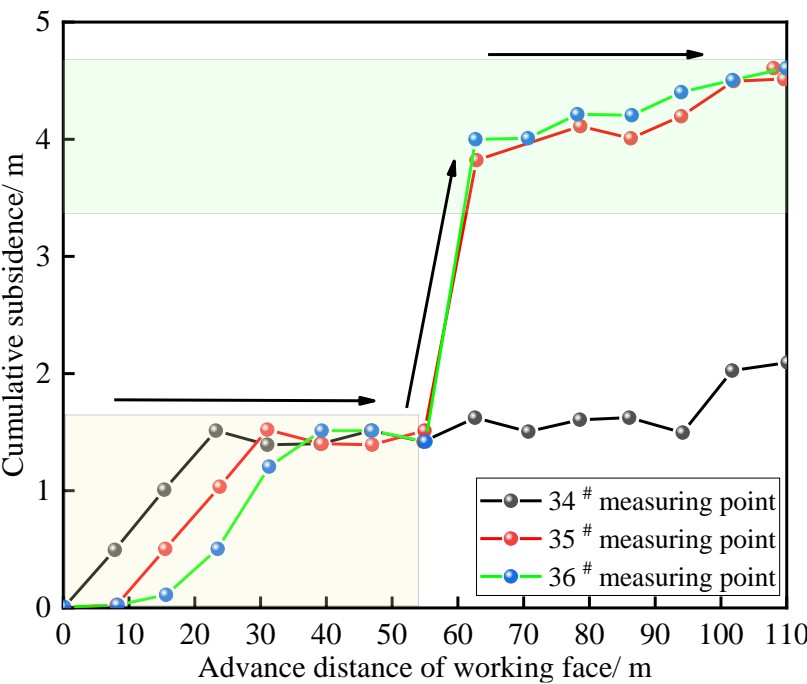

**Figure 11.** Curves of basic roof subsidence and cross fall.

### 5.3. Law of Basic Roof Strata Falling

As shown in Figure 12, the initial displacement of advancement at the working face is small, indicating that slow sinking occurs at this time, and the maximum displacement is at most 0.8 m. When the working face advances to 57.6 m, the subsidence displacement of these two measuring points increases sharply, which indicates that the rock stratum where these two measuring points are collapses synchronously with the collapse of the basic roof.

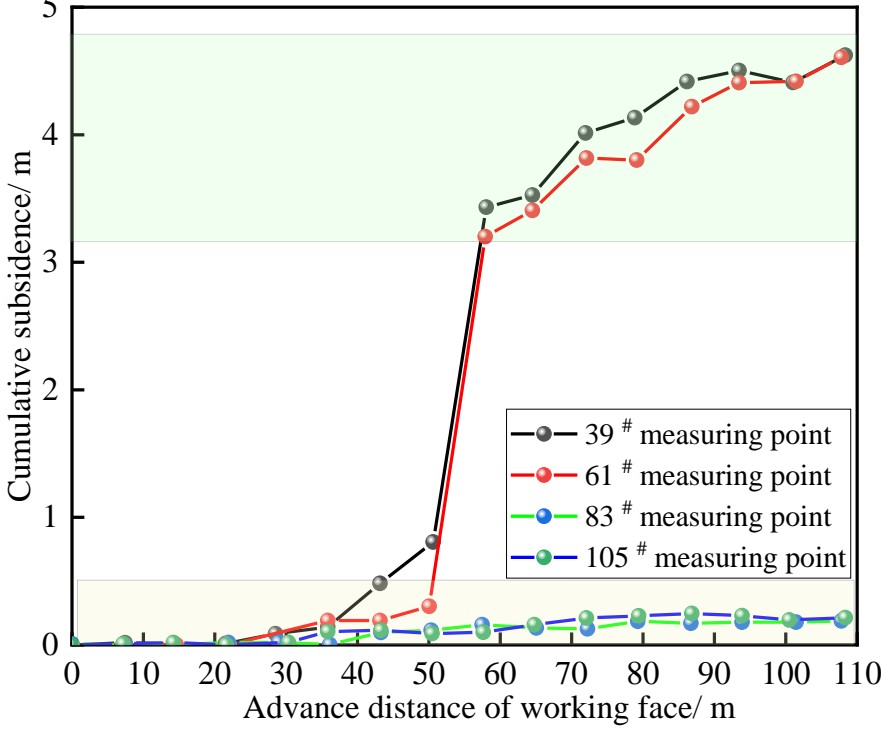

**Figure 12.** Curves of subsidence and cross fall for rock strata above the basic roof.

Measuring points 83 and 105, arranged on the upper part of the model, are not affected by the excavation of the working face. During the overall advancement of the working face, the maximum displacement of the two measuring points from beginning to end does not exceed 0.25 m. According to the analysis, in the initial mining stage, the maximum height of the fracture zone in Model II is 14.36 m, which is 3.11 times the excavation height of the stope face (4.62 m).

*5.4. Stress Distribution Characteristics of the Working Face in the Initial Mining Period*

The model is identical regarding mining conditions and stress measuring point layout, but the difference is that the mining speed is different. The model chooses the average value of the above monthly footage of 39 m for simulation. In addition, due to the influences of test site factors during excavation, the model excavation is conducted from right to left.

No. 5~7 stress sensors are arranged in the model, and the measured stress values along the advancement distance of the working face are shown in Table 3.

**Table 3.** Stress sensor test data in the first coal seam roof.

| Distance between Stress Sensor and Working Face (m) | #5 Stress Measuring Point (MPa) | #6 Stress Measuring Point (MPa) | #7 Stress Measuring Point (MPa) |
|---|---|---|---|
| 0 | 23.4 | 24.3 | 28.3 |
| 3 | 30.1 | 37.1 | 36.3 |
| 6 | 39.2 | 38.2 | 42.6 |
| 9 | 49.1 | 46.2 | 50.2 |
| 12 | 48.6 | 42.1 | 46.3 |
| 15 | 40.2 | 45.2 | 43.1 |
| 18 | 42.1 | 30.2 | 41.4 |
| 21 | 35.1 | 31.2 | 43.6 |
| 24 | 35.6 | 28.3 | 42 |
| 27 | 28.2 | 27.3 | 38.2 |
| 30 | 29.1 | 29.3 | 33.2 |
| 33 | 18.3 | 17.9 | 36.1 |
| 36 | 11.2 | 19.2 | 26.3 |
| 39 | 12.3 | 16.2 | 23 |
| 42 | | 15.6 | 13.2 |
| 45 | | 12.1 | 16.2 |
| 48 | | | 18.2 |
| 51 | | | 12.3 |

As shown in Figure 13, measuring point 5 is nearly 500 m from the surface, and its original rock stress is approximately 12.5 MPa, which is basically the same as the initial stress value (12.3 MPa) before mining. With the continuous advancement of the working face, the stress value of the No. 5 sensor increases continuously and reaches a peak value when the distance is 9 m. The maximum value is 49.1 MPa. The stress concentration factor is 3.93.

Compared with the No. 5 measuring point, the stress increase trend of the No. 6 measuring point and No. 7 measuring point comes later because the mining sequence of Model II is determined from right to left. The peak values reach the stress peak at 9 m, the maximum values are 46.2 MPa and 50.2 MPa, and the stress concentration coefficients are 3.70 and 4.02, respectively.

Among the three lines of stress measuring points arranged in the model, No. 16~18 stress sensors are set in the middle layer, and the measured stress values along with the advancement distance of the working face are shown in Table 4.

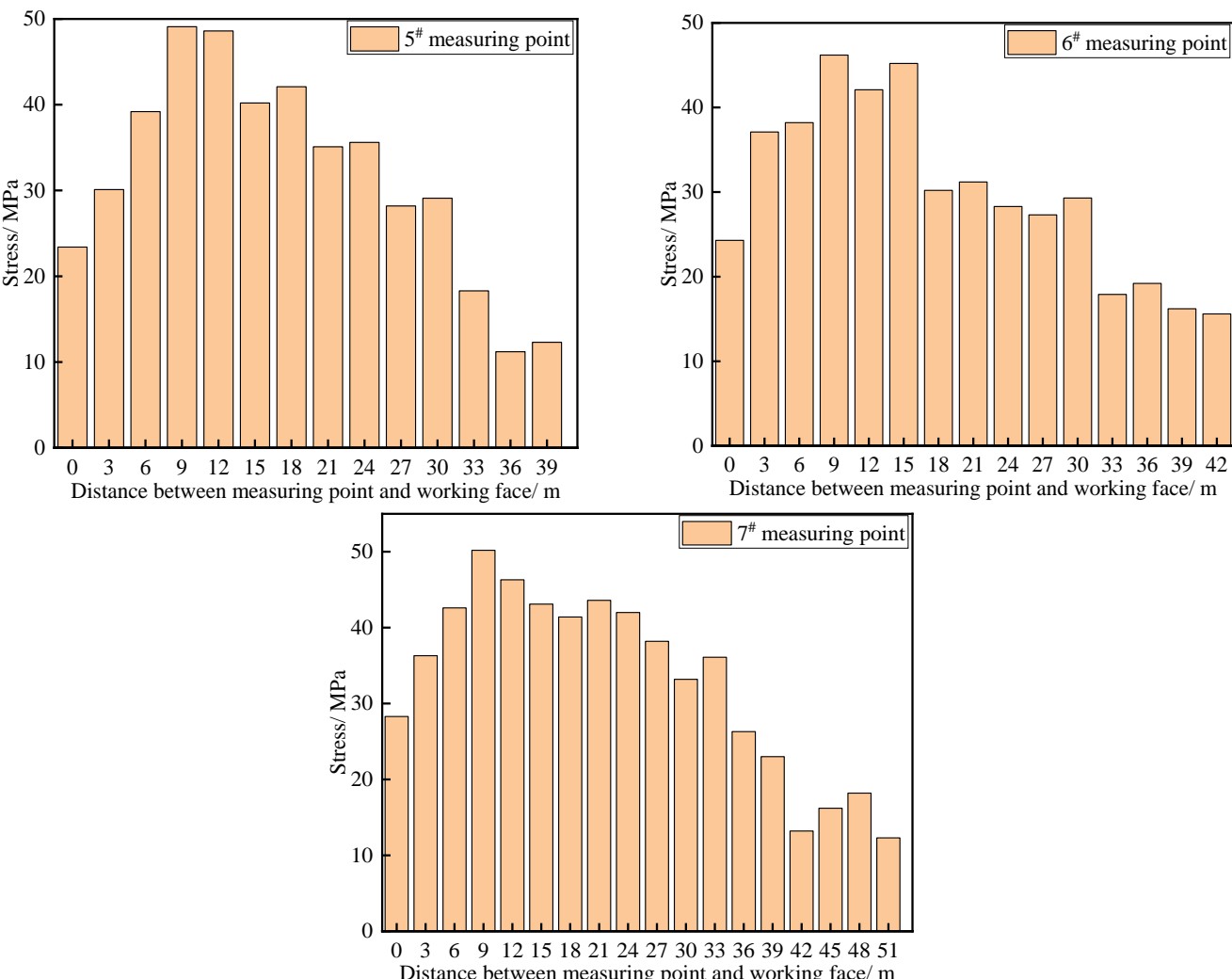

**Figure 13.** Test data curves of the first layer stress sensor in the coal seam roof.

**Table 4.** Stress sensor test data in the second coal seam roof.

| Distance between Stress Sensor and Working Face (m) | #16 Stress Measuring Point (MPa) | #17 Stress Measuring Point (MPa) | #18 Stress Measuring Point (MPa) |
|---|---|---|---|
| 0 | 24.8 | 24.3 | 22.6 |
| 3 | 23.1 | 30.1 | 28.3 |
| 6 | 26.3 | 42.6 | 35.9 |
| 9 | 32.4 | 37.2 | 21.3 |
| 12 | 35.2 | 40.2 | 22.3 |
| 15 | 27.2 | 30.4 | 19.3 |
| 18 | 29.3 | 32.6 | 22.1 |
| 21 | 28.2 | 26.3 | 15.3 |
| 24 | 23.2 | 25.9 | 16.3 |
| 27 | 24.2 | 21.3 | 12.3 |
| 30 | 19.2 | 23.2 | 13.2 |
| 33 | 19.3 | 14.3 | 10.3 |
| 36 | 15.2 | 13.2 | 15.3 |
| 39 | 11.8 | 15.2 | 13.2 |
| 42 |  | 12.2 | 14.3 |
| 45 |  | 12.5 | 10.3 |
| 48 |  |  | 13.2 |
| 51 |  |  | 11.3 |

As shown in Figure 14, the No. 16 measuring point is nearly 500 m from the surface, and its original rock stress is approximately 12.5 MPa, which is basically the same as the initial stress value measured before mining (11.8 MPa). With the continuous advancement of the working face, the stress value of the No. 16 sensor increases continuously and reaches its peak value when the distance is 12 m, reaching a maximum value of 35.2 MPa. The stress concentration factor is 2.82.

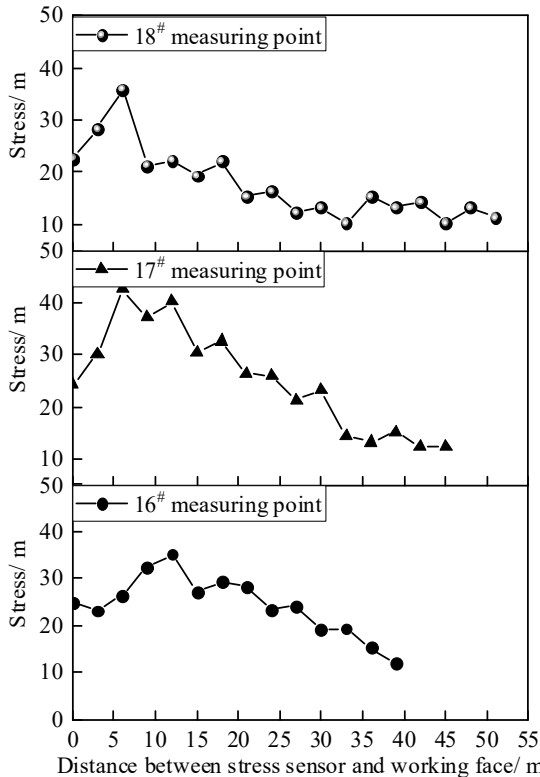

**Figure 14.** Test data curve of the second layer stress sensor in the coal seam roof.

Compared with the No. 16 measuring point, the stress of the No. 17 measuring point and No. 18 measuring point increases slightly later. This phenomenon occurs because the mining sequence of Model II is determined from right to left, and its peak values reach the stress peak at 6 m. The maximum values are 42.6 MPa and 35.9 MPa, respectively, and the stress concentration coefficients are 3.41 and 2.87, respectively.

Among the three lines of stress measuring points arranged in the model, No. 27~29 stress sensors are set in the highest floor, and the measured stress values along with the advancement distance of the working face are shown in Table 5.

**Table 5.** Stress sensor test data in the third coal seam roof.

| Distance between Stress Sensor and Working Face (m) | #27 Stress Measuring Point (MPa) | #28 Stress Measuring Point (MPa) | #29 Stress Measuring Point (MPa) |
|---|---|---|---|
| 0 | 23.7 | 20.3 | 30.2 |
| 3 | 20.9 | 21.9 | 31.6 |
| 6 | 30.6 | 25.3 | 29.3 |
| 9 | 31.2 | 31.6 | 35.2 |
| 12 | 28.3 | 30.3 | 31.2 |
| 15 | 25.6 | 25.7 | 32.6 |
| 18 | 26.3 | 23.2 | 29.6 |
| 21 | 23.4 | 26.7 | 28.6 |
| 24 | 25 | 21.3 | 26.3 |

**Table 5.** *Cont.*

| Distance between Stress Sensor and Working Face (m) | #27 Stress Measuring Point (MPa) | #28 Stress Measuring Point (MPa) | #29 Stress Measuring Point (MPa) |
|---|---|---|---|
| 27 | 20.3 | 19.6 | 23.6 |
| 30 | 21.3 | 19.9 | 24.3 |
| 33 | 22.6 | 15.3 | 19.3 |
| 36 | 15.9 | 16.5 | 13.6 |
| 39 | 10.3 | 10.9 | 18.6 |
| 42 | | 13.5 | 14.3 |
| 45 | | 12.9 | 16.3 |
| 48 | | | 13.6 |
| 51 | | | 12.9 |

As shown in Figure 15, measuring point #27 is nearly 500 m from the surface, and its original rock stress is approximately 12.5 MPa, which is basically the same as the initial stress value measured before mining (10.3 MPa). With the continuous advancement of the working face, the stress value of the No. 27 sensor increases continuously and reaches its peak value when the distance is 9 m, reaching a maximum value of 31.2 MPa. The stress concentration factor is 2.49.

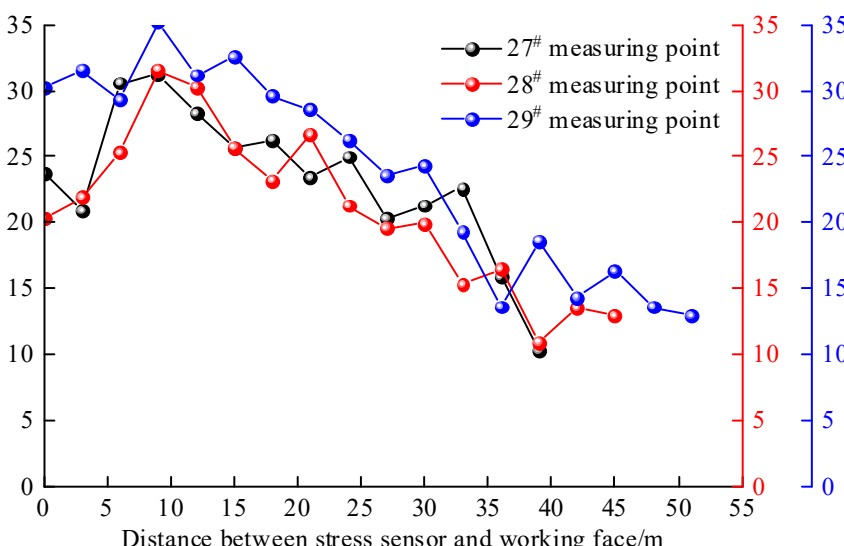

**Figure 15.** Test data curve of the third layer stress sensor in the coal seam roof.

Relative to the No. 27 measuring point, the stress of the No. 28 measuring point and the No. 29 measuring point increases slightly later because the mining sequence of Model II is determined from right to left, and its peak values reach the stress peak at 9 m, with maximum values of 31.6 MPa and 35.2 MPa, respectively, and stress concentration coefficients of 2.53 and 2.82, respectively.

### 5.5. Evolution and Distribution Characteristics of Overburden Fractures during the Initial Mining Period

Throughout the advancement of the initial mining period, the fracture evolution characteristics of the overlying strata experience three stages: slow fracture development, rapid derivation of transverse fractures and formation of vertical three zones.

Slow fracture development stage. In the model, from the working face open-off cut that advances to 26.98 m, the overlying strata are subjected to overburden load transfer and mining action, showing a migration form of bending subsidence deformation, and a small transverse parting fissure forms 2 m above the goaf. With the continuous advancement of

the working face, the phenomenon of layer separation becomes increasingly obvious, and the horizontal cracks gradually expand upwards; however, the vertical cracks have yet to appear. When the working face advances to 31.2 m, vertical cracks appear for the first time, and the interlayer transverse cracks extend and slightly bend at the direct roof rock layer 2 m above the working face. In the above stages, the mining-induced fractures show the characteristics of evolution of mining-induced fractures before the first weighting, which are relatively slow in development, relatively rare in derivation and difficult to connect with vertical fractures.

Rapid derivation stage of transverse fractures. When the working face advances to 37.2 m, under the actions of self-weight and overburden pressure, the direct roof collapses for the first time, and a large range of vertical cracks begins to form. However, with the continuous advancement of the working face, vertical through-cracks no longer develop, and the crack evolution is still dominated by rapid lateral derivation. As a result, after the first collapse of the direct roof, it does not enter the collapse stage of the basic roof and experiences four successive collapses of the direct roof.

Formation stage of vertical three zones. After the initial weighting, the mining fissures begin to enter the high-speed development stage. Each cycle of weighting is accompanied by the successive collapse of many basic roof and direct roof rocks, and many horizontal, vertical and even penetrating fractures are formed in front of the work. During mining and falling, due to the self-weight of the coal rock mass, the fallen gangue is gradually compacted, forming a falling zone and compaction area. Additionally, due to the insufficient filling of the goaf by falling debris at the initial stage of mining, the periodic separation, fracture coalescence and collapse of the overlying coal rock mass over the goaf occur continuously under the influences of the stress and mining factors, forming a stable development area and fracture zone. In addition, the main key layer has not changed significantly and only slightly and completely subsides. From the beginning to this time, three vertical zones begin to form, as shown in Figure 16.

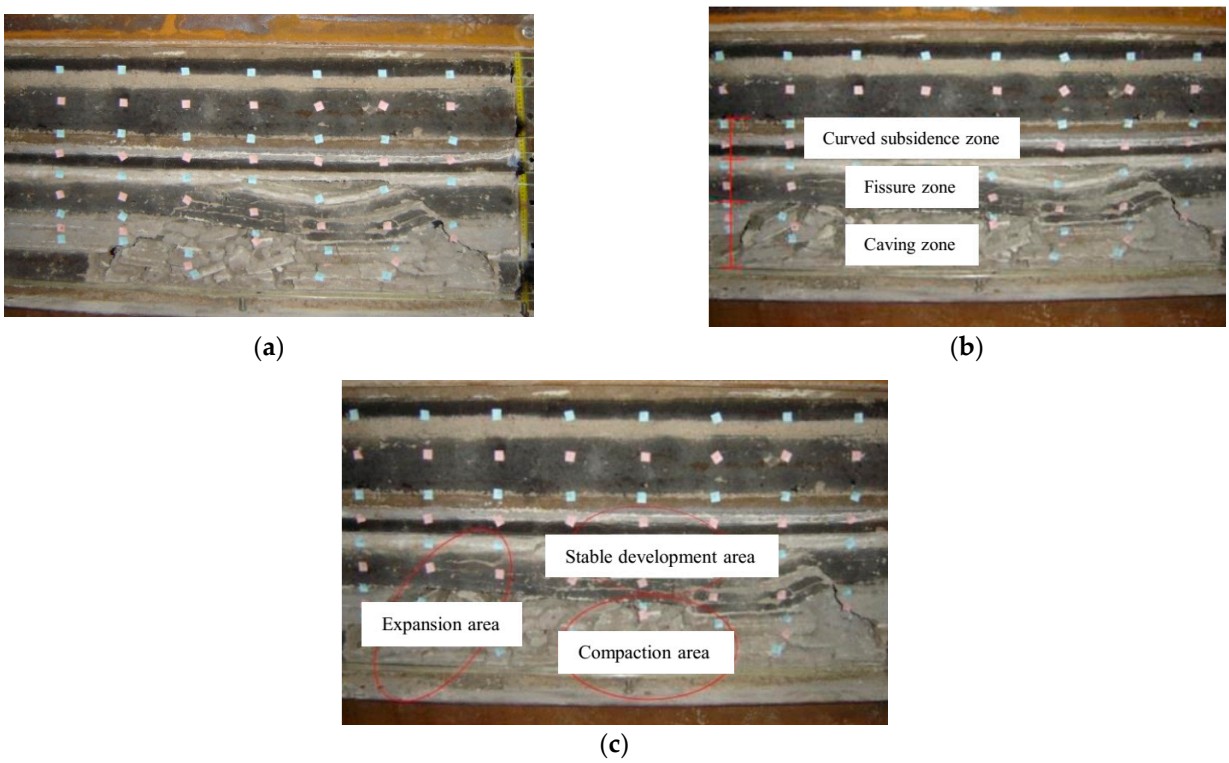

(**a**)

(**b**)

(**c**)

**Figure 16.** Evolution laws for the distribution of cracks in overburden rocks during mining. (**a**) Fracture distribution advancing to 57 m. (**b**) Three zones advancing to 68 m. (**c**) Fracture zone.

During the periodic weighting stage, with the advancement of the mining face, the pressure on the overlying strata is constantly increasing, and the scope of the damage area is gradually expanding. When the working face advances from 68 m to 83 m, the basic roof periodically collapses for the second time. The height of the parting fissure has expanded to the seventh layer (medium sandstone layer), showing a rapid development trend in length and height, as shown in Figure 17a,b. When the working face advances to 91 m, the basic roof collapses periodically for the third time. Different from the rapid development of the fracture area in the length and height directions during the previous two periodic collapses, the change in the fracture area in the height (vertical) stabilizes, and it extends rapidly in a horizontal direction; then, the fracture area increases again, as shown in Figure 17c. When the working face continues to 101 m, the basic roof periodically collapses for the fourth time. At this time, the fissure reaches 30 m in the height direction and eventually stabilizes at 6.49 times the mining height, as shown in Figure 17d.

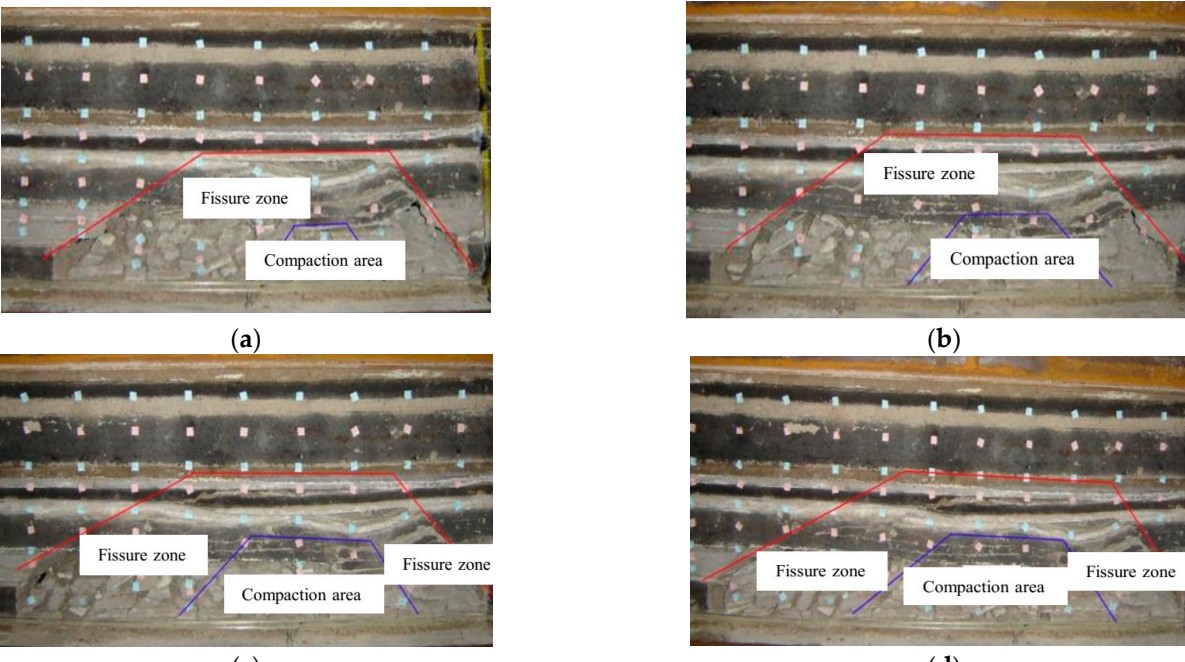

**Figure 17.** Fracture distributions at different advancement distances. (**a**) Working face advancing to 68 m. (**b**) Working face advancing to 83 m. (**c**) Working face advancing to 91 m. (**d**) Working face advancing to 101 m.

## 6. Discussion

The initial mining period is often characterized by a transitional process compared to the steady state of cyclical stresses, phase changes in displacements, and normal gas emergence during the steady mining period. During this period, the overlying rock stress transitioned from slow decompression and rapid decompression to periodic decompression, the displacement transitioned from a slow increase and rapid growth to phased changes, and the gas transitioned from desorption enrichment and abnormal outflow to balanced outflow. Load transfer and fracture evolution due to the structural deformation of key seams can induce mechanical behavioral responses such as deformation, fracture and rupture in gas-bearing coal seams, as well as altering material transport conditions such as desorption, diffusion and seepage of gas in coal seams. The development of fissures in the coal and rock has become a channel for gas transport, thus making the gas gushing from the working face larger, uneven and even inducing gas dynamic phenomena. Therefore, the specificity of the deformation pattern of the overburden during primary mining determines the stage and complexity of the gas emergence characteristics, and the study of the gas

emergence pattern can in turn be used as supporting evidence for stress transfer and fracture derivation [46,47].

The dynamic process of coal seam extraction is accompanied by mechanical behavioral responses such as deformation, decompression and even breakage of the surrounding rock body in the quarry space, which is a process of fracture evolution and the coupling of gas transport in the gas-bearing coal rock body mining, and is the root cause of abnormal gas outflow and overburden accidents in the quarry. Compared with the normal mining stage, the initial mining stage presents instability in production and gas outflow, which is the result of the interaction between overburden transport and gas flow in this stage. Compared with the stable mining phase, the initial mining phase has distinctive characteristics of coal rock deformation and gas outflow. Practice and research have shown that there is a positive correlation between the rate of recovery and the amount of unloading gas gushing out. During the initial mining phase, as the recovery rate increases, so does the amount of gas extracted. The recovery rate reflects the temporal and spatial changes in the mechanical structure of the overburden in the workings, which in turn affects the evolution of the mining fractures in the overburden and the amount of unloading gas gushing out [48–51].

During the normal mining phase, as the rate of recovery stabilizes, the amount of gas extracted and the amount of gas gushing out also stabilizes. In other words, if there is no extraction system in place, gas released from the mining-induced overburden can suddenly gush into the workings. Therefore, the instability of production and gas emergence is the main characteristic of the initial mining period, and the two are mutually influential, wherein the stage and specificity of the overburden deformation determines the dynamic instability of the evolution of the mining fissures and the abnormal emergence of the unloading gas, and even gas overload. In turn, the abnormal gas emergence makes the advancement rate of the retrieval workings unstable.

During the initial mining period, the key layer is in the process of bending deformation and sinking before the initial incoming pressure, which accounts for about 2/3 of the entire initial mining period, and the controlling role of the key layer determines that the overburden stress field during the initial mining period is dominated by tensile stress distribution. As can be seen from the previous numerical simulation results, after the initial incoming pressure, as the overburden sinking increases, the stress field of the overburden layer in the mining area gradually forms a "∩" type stress arch, but the arch height at this stage is lower than that of the stable mining period. The distribution characteristics of the stress field during the initial mining period determine the distribution pattern of the displacement field and fracture field. Before the initial incoming pressure, the overlying rock layer on the side of the extraction area is mainly distributed by horizontal fissures, and there are few vertical penetration fissures. After the initial incoming pressure in the vertical direction, the penetrating fissures develop rapidly towards the upper rock layer and basically stabilize at a certain height after the periodic incoming pressure. The overlying rock seam above the key seam in front of the workings is subject to horizontal tensile stress, and there is a "pressure relief zone" in the area approximately two times the mining height from the coal seam.

## 7. Conclusions

(1) By taking the #3 + 4 coal seam in the 24207 working face of the Shaqu Mine as the research object, the distribution laws of overburden deformation, collapse and fracture area during the initial mining period are quantitatively studied by using similar simulations.

(2) While advancing the working face from the open cut to 19 m, there is no collapse or separation. When the working face advances for an additional 3 m, the cracks enter the slow development stage. Afterwards, the number of transverse cracks in the rock stratum increases with every 3 m of working face advancement. When the working face advances to 28 m, the direct roof collapses for the first time, and the direct roof exhibits slight vertical cracks. The second, third, and fourth collapses of the direct

roof follow, indicating that the horizontal and vertical fractures of the overlying strata slowly expand and extend to the third and fourth layers.

(3) When the working face advances to 57 m, the basic roof collapses for the first time, indicating that the mining-induced fractures of the overlying strata enter a period of rapid development, the numbers of horizontal fractures and vertical fractures increase rapidly, and the scope expands to 6 layers. The working face moves forwards by 11 m again. In this process, the fracture area increases rapidly, it is highly developed and stable in the medium sandstone layer, and new fractures are constantly generated or opened.

(4) When the working face advances from 83 m to 101 m, the third and fourth periodic weighting occurs on the basic roof. The opening and closing speeds of the mining fractures are basically the same at this time, and the fracture and compaction areas stabilize.

**Author Contributions:** Software, Y.F.; Formal analysis, Y.H.; Writing—original draft, C.L.; Writing—review & editing, X.S. All authors have read and agreed to the published version of the manuscript.

**Funding:** Supported by the Scientific and Technological Innovation Programs of Higher Education Institutions in Shanxi (2021L334), by the Fundamental Research Program of Shanxi Province (202103021224277, 202203021222184), and by the Taiyuan University of Science and Technology Scientific Research Initial Funding (20222112).

**Institutional Review Board Statement:** Not applicable.

**Informed Consent Statement:** Not applicable.

**Data Availability Statement:** The data used to support the findings of this study are available from the corresponding author upon request.

**Conflicts of Interest:** The authors declare that they have no conflict of interest regarding the publication of this study.

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
