# Peer review of "Fracture Evolution Characteristics and Deformation Laws of Overlying Strata during the Initial Period of Longwall Mining: Case Study"

_sustainability, doi:10.3390/su15118596_

Round 1
Reviewer 1 Report
The following comments and suggestions in the paper should guide the authors to revise the paper.
1. The manuscript needs the following changes to reach the standard of publication: (1) improve the language, (2) improve figures, (3) improve the expression of innovation and research significance.
2. There are several language problems in the paper.
Firstly, some expressions are very awkward. For example, in the Introduction Part, “for a many years in the future” is usually expressed as “for many years to come” in English. “the position of coal as the main energy source and strategic core of China will not be shaken" could be better phrased as "coal will remain the main energy source and strategic core of China for the foreseeable future".
Secondly, “For example, the mining of coal resources collapses and destroys the overlying strata of the working face, and the overburden fissures induced by mining provide advantageous passages for water seepage, spontaneous coal combustion and abnormal gas emission in the stope, which is one of the main factors causing safety production accidents in coal mines”-Such sentences are too long, which make it hard to be understood. Please broke them in shorter sentences
Please modify the language thorough to make the expression more concise, to the pointed, and natural in English academic writing style.
3. The abstract suggests further improvement. It is suggested to modify the abstract in such a structure: question and background - research methods and objectives - results - conclusions and understanding obtained from discussion - innovation and significance.
Note that the abstract helps the readers to find out quickly whether the writing is of interest or not, so it is important to clearly indicate the innovation and new progress of the paper, or explain the differences from the past understandings.
4. Select the main research objectives, methods, etc. as keywords. Please note that the selected keywords must be consistent with the theme concept of the paper
5. Do not self-evaluate the significance of research results in the introduction. e.g. The results reveal the temporal and spatial variation relationships between the mining speed, advancement degree, and fracture evolution of the working face and surrounding rock stress. The research results provide important parameters for the position selection of grouting for the prevention of water inrush and spontaneous coal combustion and the drilling of gas drainage in the initial period of longwall mining, ensuring the safe and efficient production of the mine.
Please consider following related papers to better enhance the introduction:
[1]Cunjin Lu, Jinpeng Xu*, Qiang Li, etal. Research on the Development Law of Water-Conducting Fracture Zone in the Combined Mining of Jurassic and Carboniferous Coal Seams. Applied Sciences-Basel, 2022, 12(21): 11178.
[2]Cunjin Lu, Jinpeng Xu*. Development law of water conducting fissures in the goaf of multiple coal seams based on similar material simulation – taking the majiliang coal mine as an example. Fresenius Environmental Bulletin, 2022, 31(3): 2545-2553.
6. Figure 1: The stratigraphic column is non-standard. Suggest modifying to a professional geological histogram.
Table 2 suggests remaking with all characters centered. Why use underline?
Figure 2: There should be no characters bigger than the main text in the figure. E.g. “goaf”
Figure 5-6: Revise the layout of figures and text.
Figure 8: Please use professional software (Coredraw, PS..) to redraw. Do not use screenshot tools to create figure.
Figure 16-17: Revise the layout of the figures.
7. Please provide a brief introduction to the mining situation of Coal Seam 2 in the engineering background. Is there a goaf?
8. Is there any measured data for comparison, and how can the reliability of the model results be verified in the theoretical formula calculation section for overburden height added before the experiment?
9. The analysis of the failure law of overlying rock should follow the law of stress displacement failure. Please explain how the stress monitoring points are only arranged in the lower part and not corresponding to the range of displacement monitoring points?
10. The model mining design on page 8 should be placed in section 3.3.
11. It is suggested to add a new discussion section: discussing the significance of research understanding in the paper, including scientific significance and how it can guide practice (The purpose of this study is gas prevention and control. Please provide practical guidance for gas prevention and control work). The current paper structure is more like a research report than a paper that can attract readers. For readers who are not in this field, it is difficult to directly understand the innovation of the research from the paper.

There are several language problems in the paper.
Firstly, some expressions are very awkward. For example, in the Introduction Part, “for a many years in the future” is usually expressed as “for many years to come” in English. “the position of coal as the main energy source and strategic core of China will not be shaken" could be better phrased as "coal will remain the main energy source and strategic core of China for the foreseeable future".
Secondly, “For example, the mining of coal resources collapses and destroys the overlying strata of the working face, and the overburden fissures induced by mining provide advantageous passages for water seepage, spontaneous coal combustion and abnormal gas emission in the stope, which is one of the main factors causing safety production accidents in coal mines”-Such sentences are too long, which make it hard to be understood. Please broke them in shorter sentences
Please modify the language thorough to make the expression more concise, to the pointed, and natural in English academic writing style.
Author Response
Summary of Changes and Responses to Comments
We would like to thank the editor and reviewers for their insightful comments and suggestions on our paper. Those comments are all valuable and very helpful for revising and improving our paper, as well as the important guiding significance to our researches. We have studied comments carefully and made corrections which we hope meet with approval. Revised portion are marked in red in the paper. Below, we list the issues raised by the editor and reviewers and our responses.
Comment 1: The manuscript needs the following changes to reach the standard of publication: (1) improve the language, (2) improve figures, (3) improve the expression of innovation and research significance.
Response: Following your insightful suggestion, the language, figures and research significance have been improved.
Comment 2: There are several language problems in the paper.
Firstly, some expressions are very awkward. For example, in the Introduction Part, “for a many years in the future” is usually expressed as “for many years to come” in English. “the position of coal as the main energy source and strategic core of China will not be shaken" could be better phrased as "coal will remain the main energy source and strategic core of China for the foreseeable future".
Secondly, “For example, the mining of coal resources collapses and destroys the overlying strata of the working face, and the overburden fissures induced by mining provide advantageous passages for water seepage, spontaneous coal combustion and abnormal gas emission in the stope, which is one of the main factors causing safety production accidents in coal mines”-Such sentences are too long, which make it hard to be understood. Please broke them in shorter sentences
Please modify the language thorough to make the expression more concise, to the pointed, and natural in English academic writing style.
Response: The language has been revised to allow for a more concise, poignant and natural style of English academic writing.
Comment 3: The abstract suggests further improvement. It is suggested to modify the abstract in such a structure: question and background - research methods and objectives - results - conclusions and understanding obtained from discussion - innovation and significance.
Note that the abstract helps the readers to find out quickly whether the writing is of interest or not, so it is important to clearly indicate the innovation and new progress of the paper, or explain the differences from the past understandings.
Response: The abstract has been improved.
Comment 4: Select the main research objectives, methods, etc. as keywords. Please note that the selected keywords must be consistent with the theme concept of the paper.
Response: The keywords has been improved.
Comment 5: Do not self-evaluate the significance of research results in the introduction. e.g. The results reveal the temporal and spatial variation relationships between the mining speed, advancement degree, and fracture evolution of the working face and surrounding rock stress. The research results provide important parameters for the position selection of grouting for the prevention of water inrush and spontaneous coal combustion and the drilling of gas drainage in the initial period of longwall mining, ensuring the safe and efficient production of the mine.
Please consider following related papers to better enhance the introduction:
[1]Cunjin Lu, Jinpeng Xu*, Qiang Li, etal. Research on the Development Law of Water-Conducting Fracture Zone in the Combined Mining of Jurassic and Carboniferous Coal Seams. Applied Sciences-Basel, 2022, 12(21): 11178.
[2]Cunjin Lu, Jinpeng Xu*. Development law of water conducting fissures in the goaf of multiple coal seams based on similar material simulation – taking the majiliang coal mine as an example. Fresenius Environmental Bulletin, 2022, 31(3): 2545-2553.
Response: The significance of research results in the introduction has been deleted, and the introduction improved.
Comment 6: Figure 1: The stratigraphic column is non-standard. Suggest modifying to a professional geological histogram.
Table 2 suggests remaking with all characters centered. Why use underline?
Figure 2: There should be no characters bigger than the main text in the figure. E.g. “goaf”
Figure 5-6: Revise the layout of figures and text.
Figure 8: Please use professional software (Coredraw, PS..) to redraw. Do not use screenshot tools to create figure.
Figure 16-17: Revise the layout of the figures.
Response: Figure has been modified.
Comment 7: Please provide a brief introduction to the mining situation of Coal Seam 2 in the engineering background. Is there a goaf?
Response: The No. 2 coal seam is located in the upper part of the Shanxi Group and is the more stable and mostly mineable coal seam in the entire wellfield. The coal seam has a high gas content and the mine has a high gas gush (relative gas gush 9.10 to 26.83 m3/t, average 17.97 m3/t; absolute gas gush 15.80 to 46.58 m3/ min, average 31.18 m3/ min). The lithology surrounding Coal Seam No. 2 is mainly carbonaceous mudstone, mudstone and sandy mudstone, followed by fine-grained sandstone and siltstone. After the mining of the working face, the free caving method is used to treat the goaf, and the untreated goaf is located behind the working face. The mudstone is generally thick (10.12-27.18 m, average 16.67 m), generally good integrity and dense and hard. The ability of the surrounding rocks of the No. 2 coal seam to cap the gas is good and the gas is easily preserved and enriched for reservoir formation, which is one of the main geological factors contributing to the generally high gas content of the seam.
Comment 8: Is there any measured data for comparison, and how can the reliability of the model results be verified in the theoretical formula calculation section for overburden height added before the experiment?
Response: The field measured data is not analyzed. The similar simulation in this study is to explore the relationship between overburden fractures and gas emission. Field measurement will be studied later to verify the reliability of the model results.
Comment 9: The analysis of the failure law of overlying rock should follow the law of stress displacement failure. Please explain how the stress monitoring points are only arranged in the lower part and not corresponding to the range of displacement monitoring points?
Response: Stress monitoring uses monitoring equipment such as pressure boxes, which need to be pre-built into the model in advance, causing errors in displacement monitoring. In order to make the displacement monitoring point data more accurate during the experiment, this test used the displacement and stress separation method.
Comment 10: The model mining design on page 8 should be placed in section 3.3
Response: The model mining design has been improved.
Comment 11: It is suggested to add a new discussion section: discussing the significance of research understanding in the paper, including scientific significance and how it can guide practice (The purpose of this study is gas prevention and control. lease provide practical guidance for gas prevention and control work). The current paper structure is more like a research report than a paper that can attract readers. For readers who are not in this field, it is difficult to directly understand the innovation of the research from the paper.
Response: New discussion section has been added.
Thank you for your comments. The manuscript has revised by professionals.
Special thanks to you for your helpful comments.
We tried our best to improve the manuscript and made changes as needed. These changes will not influence the content and framework of the paper.
We earnestly appreciate for editor’s/reviewers’ warm work, and hope that the correction will meet with approval.
Once again, thank you very much for your comments and suggestions.

Reviewer 2 Report
Dear Authors,
The manuscript "Study on the Fracture Evolution Characteristics and Deformation Laws of Overlying Strata During the Initial Period of Longwall Mining" presents an interesting research problem. It concerns, a specific case, i.e. the working longwall 24207 of the Shaqu mine. However, this research is not universal to other cases (other rocks, other thicknesses, etc.) than the analyzed object. Therefore, "case study" should be added to the title of the article. I do not belittle the importance of your studies. They are a great help to the Shaqu mine exploiting longwall 24207, in the context of uncontrolled outflow of methane. I have carried the rest of my comments and remarks in the manuscript.

Author Response
We have carefully considered all the suggestions from the editor and reviewers and modified the paper accordingly. Our point-by-point responses in the manuscript .

Reviewer 3 Report
The manuscript describes the process of investigating anomalous gas release during the initial stage of development of long drifts. The authors analyzed various advance rates and their effects on rock fracture deformation and distribution using a simplified mechanical model of the rock structure. The results showed that the evolution of fractures goes through slow and fast stages of growth. The range of influence of weighting step distance and overburden separation of the coal wall is related to the advance rate, and the proper increase in the advance rate of the face helps to prevent the occurrence of abnormal gas emissions.
The results described are important for practical applications in the mining industry and can be used to optimize the development of long drifts. The text is written in an accessible and understandable style, using specialized terminology. Overall, this material is useful and interesting for specialists in the field of mining and can be published in the journal.
The authors did a good review of the literature in the Introduction and outlined the main focuses of its research. In my opinion, another 5-7 new papers on this topic should be added (the list of references is too short, it would be better to expand it).
In general, the results are consistent with each other and well presented in the main conclusions of the investigation. The article with minor corrections can be recommended for publication.
Minor comments:
1. Format the entire manuscript according to journal requirements
2. It is desirable to expand the list of references a little by adding fresh articles on the topic of research. For international journals, the unofficial minimum is 50 references.
3. Figures 4, 5 are small print, please enlarge.
Author Response
Comment 1: Format the entire manuscript according to journal requirements.
Response:. I have improved the entire manuscript according to journal requirements.
Comment 2: It is desirable to expand the list of references a little by adding fresh articles on the topic of research. For international journals, the unofficial minimum is 50 references.
Response: References have been increased to 51.
Comment 3: Figures 4, 5 are small print, please enlarge.
Response: Figures 4 and 5 have been modified.

Reviewer 4 Report
The article, "The study of the characteristics of the evolution of cracks and the laws of deformation of overlying layers in the initial period of the development of long trunks" is aimed at the study of heterogeneous bodies with a peculiarity of location and genetic parameters. It is worth noting that a special individual approach is needed here, which was used by the authors of the study. That is why the results obtained differ from the general accepted patterns of crack development in such materials. The results obtained are beyond doubt. Further study of natural conditions can reveal the general patterns of the development of foci of destruction in such special objects. The article can be published.
Author Response
We appreciate the effort of the editor and reviewers and hope that the corrections are acceptable.
Once again, we appreciate your comments and suggestions.
Round 2
Reviewer 1 Report
The conclusion section is suggested to be written as follows:
By taking the #3+4 coal seam in the 24207 working face of the Shaqu Mine as the research object, the distribution laws of overburden deformation, collapse and fracture area during the initial mining period are quantitatively studied by using similar simulations.
(1) While advancing the working face from the open cut to 19 m, there is no collapse or separation. When the working face advances for an additional 3 m, the cracks enter the slow development stage. Afterwards, the number of transverse cracks in the rock stratum increases every 3 m of working face advancement. When the working face advances to 28 m, the direct roof collapses for the first time, and the direct roof slightly exhibits vertical cracks. The second, third, and fourth collapses of the direct roof follow, indicating that the horizontal and vertical fractures of the overlying strata slowly expand and extend to the third and fourth layers.
(2) When the working face advances to 57 m, the basic roof collapses for the first time, indicating that the mining-induced fractures of the overlying strata enter a period of rapid development, the numbers of horizontal fractures and vertical fractures increase rapidly, and the scope expands to 6 layers. The working face moves forwards 11 m again. In this process, the fracture area increases rapidly, it is highly developed and stable in the medium sandstone layer, and new fractures are constantly generated or opened.
(3) When the working face advances from 83 m to 101 m, the third and fourth periodic weighting occurs on the basic roof. The opening and closing speeds of mining fractures are basically the same at this time, and the fracture and compaction areas stabilize.
Minor editing of English language required